# Web-Based Assessment of the Phenomenology of Autobiographical Memories in Young and Older Adults

**DOI:** 10.3390/brainsci11050660

**Published:** 2021-05-18

**Authors:** Manila Vannucci, Carlo Chiorri, Laura Favilli

**Affiliations:** 1Department of Neurofarba, Section of Psychology, University of Florence, 50135 Firenze, Italy; laura.favilli1@stud.unifi.it; 2Department of Educational Sciences, University of Genoa, 16126 Genova, Italy; carlo.chiorri@unige.it

**Keywords:** phenomenology, autobiographical memory, web-based assessment, aging

## Abstract

Autobiographical memories (ABMs) produce rich phenomenological experiences. Although few standardized and comprehensive measures of the phenomenology of ABMs have been developed, a web-based assessment of the full range of phenomenological properties is still missing. In the present study, we aimed to fill this gap and tested the psychometric properties of a web-based version of the Assessment of the Phenomenology of Autobiographical Memory (APAM) in a group of young and older adults. Specifically, taking advantage of the flexibility of web-based assessment methodology, we tested the rating consistency of APAM items, asking participants to rate the phenomenology of their ABMs with respect to seven cues, administered in one per day in seven different days. In each session, we also collected ratings of mood and arousal. Using linear mixed modeling (LMM), we could examine whether the phenomenology ratings differed with respect to age group while controlling for sex, age of the memory, arousal, mood, and specificity of the memory. Results revealed an adequate level of consistency of ratings in both young and older adults. Moreover, LMMs revealed a more intense experience of recollection and reliving (i.e., sensory and emotional) and a higher confidence in memory accuracy in older compared to younger adults. The theoretical and practical usefulness of a web-based assessment of the phenomenology of ABMs are discussed.

## 1. Introduction

A defining feature of autobiographical memories (hereafter ABMs) is the phenomenology, that is, the subjective experience associated with remembering. The phenomenological structure of memories includes a broad set of characteristics, namely, sensory, semantic, contextual, and emotional features experienced while remembering an event, and it gives rise to the feeling of reexperiencing the past (*autonoetic* awareness), which is the hallmark of ABMs [1,2,3].

As we will review below, the investigation of the phenomenological properties of ABMs has provided an important contribution to our understanding of the differences across different types of ABMs and revealed interesting changes of ABMs associated with aging. The phenomenological profile of ABMs is also associated with individual differences in some dimensions of psychological functioning, thereby suggesting that the phenomenology of ABMs can be considered a consistent and stable characteristic, similarly to, e.g., personality traits.

Over the last years, both researchers and participants have become increasingly familiar with the Internet and self-administered web-based versions of questionnaires have become a common alternative to paper-and-pencil versions. However, to the best of our knowledge, the few psychometrically sound measures of the phenomenology of ABMs recently developed consist of paper-and-pencil questionnaires and no investigation of the psychometric properties of web-based versions of these measures has been carried out [4,5]. In the present study, we aimed to overcome this limitation and develop a web-based version of the Assessment of the Phenomenology of Autobiographical memory (APAM) [6] and test it with a sample of young and older adults.

### 1.1. Relevance of the Phenomenology of ABMs

Independent of memory content, the phenomenology of ABMs reflects both the characteristics of different types of memories and the individual’s psychological functioning. In this regard, direct comparisons between self-defining memories and earliest childhood memories revealed clear differences in the phenomenology across the two types of ABMs and suggest that the phenomenological profile of different types of ABMs may reflect their different roles for the current self and identity [7,8].

For example, Montebarocci et al. [7] found that self-defining memories were rated as more vivid, coherent, and rich in sensory details compared to earliest childhood memories, and they were experienced as more emotionally intense, more likely to be seen from a first-person visual perspective, and more likely to be shared with other people. On the contrary, earliest childhood memories were evaluated as more psychologically distant from the participants’ current self and they were less likely to be shared with others.

Age of memory (or recency) was also found to affect the phenomenology of ABMs, with recent memories rated as more vivid and emotionally intense compared to remote memories [5,8], and more frequently reported to be experienced as seen from a first-person visual perspective [9]. Recently, researchers have also explicitly looked at differences in the phenomenological experience of ABMs associated with participant age [5,10,11,12,13,14]. Globally, the comparisons between groups of young and older participants revealed a complex picture. On the one hand, older adults are more likely to recall general than specific past events with less objective sensory details [15,16,17,18,19]. On the other, older adults may experience a stronger memory phenomenology than young adults [5,10,11,12,13,14]. For example, in the study by Kingo et al. [10], older participants reported their earliest memories to be significantly more vivid and less fragmented (more coherent) than did younger participants. Similarly, in the study by Rubin and Schulkind [12], older adults reported a higher rating of vividness for important as well as word-cued memories compared to younger adults. According to the results of some studies, these differences in the phenomenology of memories may occur especially when recalling personally meaningful ABMs (i.e., ABMs that are critical in the construction of individuals’ life stories) [14]. For example, in a study carried out on a sample of 281 individuals between the ages of 18 and 94, Siedlecki et al. [14] found that participants’ age correlated with increased vividness, coherence, sensory detail, time clarity, and a first-person perspective for the personally meaningful memory but not for everyday memories.

Luchetti and Sutin [5] investigated the phenomenology of turning point memories (i.e., episodes that marked an important change in life story) in a large age-stratified sample of participants. In the study, middle-aged and older participants (i.e., 40–49, 50–59, and ≥60 years) rated their memories as more vivid, coherent, easily accessible, emotionally intense, with more sensory details and a clear time perspective than the youngest group (i.e., 20–29 years). Furthermore, they shared them more often and viewed them from a first-person visual perspective more than younger participants.

In healthy young adults, the subjective experience of ABMs is also modulated by individual differences in psychological functioning, such as personality [20,21,22], emotion regulation [23], self-esteem [8,24], and visual imagery [6]. As for self-esteem, some studies have shown that high self-esteem individuals remembered experiences more positively [8,24] and they rated their memories as more coherent compared to individuals with low levels of self-esteem [8]. 

As for visual imagery, a recent study by Vannucci, Chiorri, and Marchetti [6] has shown that higher levels of visual object imagery were associated with more sensory details and recollective qualities of memory, and with stronger experience of sensory and emotional reliving. Globally, these studies on individual differences suggested that when ABMs are triggered by a standardized set of cues, their phenomenology shows a characteristic, consistent, and stable pattern within the individual.

In this regard, Luchetti, Rossi, Montebarocci, and Sutin [4] have tested the continuity of autobiographical memory phenomenology across memories and over time (i.e., 4 week period) and they could show a moderate stability in phenomenology ratings over time (median correlation > 0.40), irrespective of memory contents, and especially for dimensions such as emotional intensity and visual perspective. In a similar vein, recently Berntsen, Hoyle and Rubin [25] developed a new measure of individual differences in autobiographical remembering, the Autobiographical Recollection Test (ART), and found a high stability (overall correlations over > 0.70) of seven aspects of recollecting ABMs over time (a delay between one and eight weeks).

However, most of these studies were carried out on undergraduates, hence it could not be tested whether this stability persisted across the life span.

### 1.2. Assessment of the Phenomenology of ABMs

Despite the relevance of the phenomenology of ABMs, for a long time, phenomenological properties have been measured using ad hoc items and the assessment has been limited to a few dimensions (e.g., vividness or rehearsal). During the last twenty years, only a few standardized and more comprehensive measures of the phenomenology of ABMs have been developed. The first tool of this sort was the Autobiographical Memory Questionnaire (AMQ) [26,27]. It comprises a set of around 20 questions (depending on the version) about the memory elicited by a set of cue words (30 in the seminal paper), and each phenomenological dimension is measured by a single item. The participant’s score on this dimension is computed averaging the ratings on the same item across all cues.

A different approach, focused on multi-item measures has been adopted in the development of other instruments, such as the Memory Experiences Questionnaire (MEQ) [8] and its shortened version [28], and the more recent Autobiographical Memory Characteristic Questionnaire (AMCQ) [29], that also extended the range of dimensions assessed by the instrument.

Multi-item measures of the phenomenological dimensions have advantages and limitations compared to single-item ones. On the one hand, they maximize the reliability of the scores for a single cue, but their supporting evidence is somehow limited to the specific cues on which they were validated (i.e., general and early memories for the MEQ, and childhood memories, autobiographical memories related to romantic relationships, and self-defining memories for the AMCQ). When one wants to assess the phenomenology as a stable disposition of the individual, to investigate its association with other dispositions or traits, it is preferable to assess the phenomenological characteristics of ABMs on (relatively) larger and heterogeneous sets of cues and assess the consistency (i.e., the stability) of the ratings of each characteristic across cues. In other words, if a phenomenological characteristic of a memory, such as, e.g., a high level of visual details, is a stable feature of an individual’s ABM, its ratings should be consistently high across the ABMs triggered by different cues, and not only for a specific memory.

Very recently, Vannucci et al. [6] developed a new measure, the Assessment of the Phenomenology of Autobiographical Memory (APAM), that enables the assessment of the phenomenology of ABMs using this approach. In developing APAM, the authors followed the Rubin et al.’s [26] approach of asking participants to think of a personal memory in response to each of twelve cue words and rate the memory on each of the 27 items tapping into a range of phenomenological properties that is larger and more comprehensive than the one assessed by the MEQ and the AMCQ, while limiting to 5–8 min the administration time. APAM items showed adequate levels of (internal) consistency and unidimensionality across the cues, thus suggesting that the phenomenology of ABMs can be considered as a stable characteristic. Moreover, item ratings (i.e., scores) were associated with other stable characteristics such as individual differences in visual object imagery.

The characteristics of APAM make it a useful and versatile instrument suitable for the assessment of the phenomenology of ABMs in a wide range of contexts, including older adults and groups of neuropsychological and clinical patients. The ease of administration of APAM also makes it a good candidate for developing a web-based assessment of the full range of phenomenological properties of ABMs, thus making the case for the present study.

During the last years, an increasing number of studies have shown clear advantages and potential of assessing cognition with online self-administered assessment tools, for both research and clinical practice [30]. For example, the use of web-based cognitive tools may increase the availability of testing and widen participation, thereby enabling the recruitment of a large number of participants and in different settings, even at home. Online-assessment can be conducted in an asynchronous mode, without direct contact with the researcher/clinician, so that the scheduling of cognitive tests can be more flexible and conducive to participants’ availability, and it may also help in reducing costs and time for both researchers/clinicians and participants/patients [31,32].

Online assessment as well as computerized tests have also other advantages over traditional testing related to the accurate recording of responses, automated scoring, and data exporting and accessibility [33], which eliminate human error and save cost and time.

Interestingly, web-based cognitive tests have been found to be promising in measuring cognition not only in healthy young adults but also in older adults [34] and in detecting mild cognitive impairment (MCI) and dementia [30,35,36]. In fact, despite the concern associated with the use of computer-based tests by older adults, due to their presumed lack of familiarity with using a computer and with the related presence of computer anxiety, some studies have shown that older adults evaluated computer-based and web-based self-administered measures more favourably than paper-and-pencil versions, and they perceived these modalities as more user-friendly, acceptable, and satisfying, less-stressful and less inducive of high levels of anxiety [37,38].

Despite the clear merits of web-based assessment, this is still a new development, and it comes with a series of limitations and challenges. For example, the retest reliability of web-based cognitive tests as well as their correlation with traditional paper-and-pencil tests varies across different instruments [34,36,39], and with more complex tests the performance can vary depending on test devices [34,40,41].

From a methodological point of view, sampling and generalizability issues associated with web-based assessment need to be carefully considered. On the one hand, the results of some studies [42] suggest that this modality allows the recruitment of samples that are not only large but also more diverse than traditionally recruited samples in terms of several background variables (e.g., geographical location, socioeconomic status). Moreover, it facilitates the access to participants in remote areas, or with physical and mobility problems. On the other hand, and especially with a sample of older adults, sampling biases may be introduced [43,44]. Older participants typically show lower rates of adoption and use of newer information and communication technologies than younger participants. As a result, any sample of older adults that possess a device connected to the Internet and are sufficiently comfortable with using it, not only for survey administration, may not be representative of the population of older adults. As reported by Quinn [44], older participants also show more concern for privacy in the online environment than younger ones, and this may lead to substantially different levels of participation and non-response.

In a similar vein, differences in contexts and settings (e.g., differences in location, timing, absence of examiner, likelihood of participant distraction) in unsupervised web-based assessments that may interact with task characteristics and influence performance would need to be carefully considered in studies. Therefore, a systematic investigation of the comparability of results/performance on the same task in unsupervised, web-based assessments and in-person paper-and-pencil assessment needs to be carried out [33,45,46].

## 2. The Present Study

In the seminal study on APAM, participants completed the questionnaire in its paper-and-pencil version and were undergraduate students ranging in age from 18 to 32. As mentioned above, the ratings on the 27 items of APAM showed levels of consistency (indexed as Cronbach’s alpha of ratings of the same item on the twelve different cue conditions) that supported the claim that the phenomenology of the ABMs is a stable disposition of the individual. The aims of the present study were (i) to seek and replicate this result in a new sample of undergraduate students; (ii) to investigate whether this result could be replicated also in a sample of older adults (aged 60–75), and (iii) whether there were differences in consistency indices between young and older participants. To perform this study, we developed a web-version of APAM that provided us the opportunity to administer it in sessions that took place in different days. Vannucci et al. [6] asked participants to rate the phenomenology of their ABMs with respect to twelve cues administered in the *same* session (day). Although the results supported the relative stability of the scores across cues *within* a single session/day, it was not possible to provide evidence for their stability also *across* sessions/days, thus accounting, e.g., for differences in mood and arousal levels. We thus designed the study to collect ratings of the phenomenology of ABMs in response to seven cue words, administered one per day in seven different sessions. This allowed us to address a further aim, namely, (iv) whether the ratings of the phenomenological characteristics of the ABMs differed with respect to age group while controlling for sex, age of the memory, arousal, mood, and specificity of the memory.

## 3. Materials and Methods

### 3.1. Participants

Potential participants were 100 students from psychology courses at the “Universita dell’Età Libera” of Florence, which enrolls older adults as students (age > 60), and 130 undergraduates of Psychology and Dietetics, enrolled in courses at the bachelor level at the University of Florence. They were invited to take part in the study through an advertisement sent to the mailing list of the course they were attending. Forty-six and 36 participants volunteered in the two groups, respectively, but one participant in each group dropped out from the study before completing it. The final group of participants comprised 45 older adults (mean age = 66.76 years, standard deviation (*SD*) = 4.08, range 60–75; females = 78%; mean years of education = 15.80, *SD* = 3.03), and 35 young adults (mean age = 22.69 years, standard deviation (*SD*) = 3.03, range 19–30; females = 83%; mean years of education = 14.06, *SD* = 2.11), all native Italian speakers.

All participants volunteered to participate after being presented with a detailed description of the procedure, and all were treated in accordance with the *Ethical Principles of Psychologists and Code of Conduct* [47]. The research has been complied in accordance with the tenets of the Helsinki Declaration [48] and has been approved by the authors’ institutional research committee. To be included in the study, participants had to report to be at least 18 years old. They did not receive any compensation for their participation.

### 3.2. Materials

Background form. In the first day only, participants were asked to complete a form that comprised questions about sex, age, and education. They were also prompted to generate their personal ID answering to some questions (not reported here to ensure confidentiality). The same questions were asked in all subsequent sessions to reproduce the same ID, which was used to match the data of different sessions.

Assessment of emotional states*:* Prior to the experimental task (see below), participants were asked to report their level of arousal (low/very calm-high/very agitated) and their mood (very negative-very positive) using a 7-point rating scale.

Assessment of the Phenomenology of Autobiographical Memory (APAM). The APAM was used to measure the phenomenological properties of autobiographical memories. Specifically, this 27-item measure assesses the following properties: clarity of memory (item 01 in APAM), color (item 02), vividness (item 03), richness of visual details (item 04), sound (item 05), smell (item 06), touch (item 07), taste (item 08), sensory reliving (item 09), auditory reliving (item 10), visual reliving (item 11), spatial reliving of the event (item 12), remember/know (item 13), formulation in words (item 14), coherence (item 15), confidence in the accuracy of the memory/testify (item 16), accessibility (item 17), visual perspective-third person (item 18), emotional intensity (event–item 19; memory–item 21), emotional reliving of the event (item 20), self-distancing (item 22), rehearsal of memories (talking-item 23), personal importance (item 24), imagined/real (item 25), specificity (item 26), and age of memory (item 27). The first 25 items were rated on a seven-point Likert-type scale. Item 26 included three choices: whether the memory was for an event that occurred once within a single day and place (item 26a), whether it was a summary or merging of similar events (item 26b), or whether it was for an event that extended for a period greater than one day (item 26c). Item 27 asked to report the age of memory on a 10-point scale from “one week” to “more than 50 years” (see Appendix B for details).

In the context of APAM, participants were asked to retrieve an autobiographical memory associated with each of the seven cue words, one per day. The cue words were all neutral and concrete (i.e., city, dress, sea, wine, mountain, plant, fire). The cue words were taken from the set used in Vannucci et al. [6], and they were all words familiar to Italian speakers.

### 3.3. Procedure

Participants were tested online. Each day they received an e-mail with a link to a web page. We used the Limesurvey platform for designing and administering the survey. The questions were administered in the following sequence: (i) Text describing the study and its aims; (ii) Informed consent (participants had to select “I accept the terms and conditions of the study” to continue); (iii) Question about whether it was the first session the participant took. (If yes) (iv) Questions for generating the personal ID code and about background information; (If no) (iv) Questions for generating the personal ID code only; (v) Questions about arousal and mood; (vi) Instructions about the task; (vii) Cue; (viii) Question about whether a memory had been triggered by the cue; (If yes) (ix) APAM items; (If no) (ix) End of the survey; (x) Question about the attention paid to the questions and comments.

The task was self-paced, and its completion took between 10 and 15 min. All participants completed the seven sessions in 7–11 days. The self-administered online version of APAM required minimal computer proficiency and all participants reported to be familiar with computer, e-mail, and online learning platforms.

## 4. Results

Data were processed and analyzed using *R* 3.5.3 [49]. A complete list of the packages used is reported in the Appendix A.

The two groups did not differ with respect to the proportion of female participants (Yates’ corrected *Z* = 0.28, *p* = 0.778, *d* = 0.13 [−0.31; 0.57]) but there was a substantial difference in education (Welch’s *t*-test: *t* (77.16) = 3.02, *p* = 0.003, *d* = 0.67 [0.20; 1.13]).

The median number of memories elicited by the seven cue words in single participants was 6 in the older group (range 4–7) and 7 in the young group (range 3–7) and did not differ between groups (Mann–Whitney test: *z* = 0.315, *p* = 0.757, effect size *r* = 04 (−0.19; 0.25). The median proportion of memories elicited by cue words was 0.90 (range 0.76–0.97), and did not differ between participants (*X*^2^ (1) = 3.08, *p* = 0.080, Cramér’s *V* = 0.08 (0.01; 0.16)).

Descriptive statistics and consistency of ratings are reported in Table 1. The consistency of ratings computed for each item across cue words, computed as Intraclass Correlation Coefficient (two-way mixed model, average measure, consistency; [50]), ranged from 0.62 to 0.94 (median 0.79) in the young group, and from 0.62 to 0.87 (median: 0.74) in the older group (see Appendix A). These analyses allowed us to address aims (i) and (ii). Aim (iii) was addressed by comparing the coefficients across groups using the procedure described in Charter and Feld [51]. After adjusting raw *p*-values for false discovery rate using the Benjamini-Hochberg procedure [52], a significant difference in ICC, albeit with a small effect size, was found for item 14 (*F* (34, 44) = 3.07, *p* = 0.001, adjusted *p* = 0.025, *r* = 0.28 (0.10; 0.48); higher ICC in young participants). The comparison for item 13 matched the significance threshold (*F* (44, 34) = 2.68, *p* = 0.004, adjusted *p* = 0.050, *r* = 0.27 (0.03; 0.49); higher ICC in older adults).

We then addressed aim (iv), i.e., whether each APAM rating, considered as an outcome variable, could be predicted by age group, sex, age of the memory, arousal, mood, and specificity of the memory, which were modeled as fixed effects. To take into account the nesting of observations in participants and in cues, these were specified as random effects in a cross-classified LMM (intercept-only model).

However, before specifying the LMMs, we performed some preliminary analyses to test whether the two groups of participants differed with respect to some theoretically relevant variables that might confound the comparisons. Since some studies have shown that differences in the phenomenology of ABMs were more apparent when recalling personally meaningful memories, which are critical in the construction of individuals’ life stories, we considered the answers to item 24 (“This memory is significant for my life because it imparts an important message for me or represents an anchor, a critical juncture, or a turning point”). Had we found differences in the personal relevance of the memory between our groups of participants, any difference in the other APAM item scores could have been explained by this difference. We thus used a cross-classified LMM [54] that took into account the nesting of observations in participants and cue words to test differences between elder and young participants on the scores of item 24. Results revealed no significant difference (Older adults Estimated Marginal Mean (EMM) (Standard Error): 3.95 (0.24); Young EMM: 3.49 (0.26); *t* (75.81) = 1.91, *p* = 0.060, *d* = 0.25 (−0.19; 0.69)). As a further check, we tested whether the proportion of 6 or 7 scores on item 24 (i.e., scores that revealed the highest personal relevance of the memory) differed between groups, and again we did not find evidence of a difference (Older adults: 0.24; Young: 0.18; Yates’ corrected *z* = 1.54, *p* = 0.123, *d* = 0.15 (−0.29; 0.60)).

Second, we tested whether there were significant group differences in the specificity of the memory (item 26). We did not find evidence of a substantial difference (*X*^2^ (2) = 1.96, *p* = 0.374, Cramér’s *V* = 0.06 (−0.02; 0.12)).

Third, we tested group differences in age of the memory, using a cross-classified LMM. In this case, the difference was statistically significant (Older EMM: 0.12 (0.06); Young EMM: −0.84 (0.07); *t* (82.00) = 11.38, *p* < 0.001, *d* = 2.53 (1.93; 3.12); to address the ordinal nature of these data, the values of age of memory were first converted into ranks and then translated into z-scores using the inverse-normal cumulative distribution function, as suggested by Gelman [55]).

Given that we had found that groups were unbalanced for educational level, too, to get correct estimates of the age group differences, we used propensity score analysis [56] to get weights to be applied to observations in subsequent LMMs that balanced the two age groups with respect to age of memory and education. Since we had to take into account the nesting of observations in participants and in cues, we used the procedure described in Cannas and Arpino [57] for multilevel cases and, after matching, the standardized differences (in Cohen’s *d* metric) were 0.07 and 0.16 for age of the memory and education, respectively, indicating an acceptable balance. The results of the subsequent cross-classified LMMs for each item are reported in Appendix A.

All other predictors kept constant, older adults endorsed significantly higher scores in clarity (item 01), vividness (item 03), sensory reliving (item 09), auditory reliving (item 10), spatial reliving of the event (item 12), remembering/knowing (item 13), formulation in words (item 14), coherence (item 15), confidence in the accuracy of the memory/testify (item 16), emotional reliving of the event (item 20), and imagined/real (item 25). In all these cases, the effect sizes ranged from moderate to strong. 

The effect of sex and positive mood was never significant, while there was a general tendency of APAM scores to decrease with age of memory, except for visual perspective/third person (item 18), self-distancing (item 02), for which scores significantly increased with age of memory. The effect sizes of the significant age of memory effects ranged from weak to moderate. Higher levels of arousal weakly predicted lower scores on clarity (item 01), vividness (item 03), visual reliving (item 11), and remembering/knowing (item 13).

The effect of specificity was significant in five items. For coherence (item 15), Single memories showed higher scores than Multiple memories. For accessibility (item 17), emotional intensity/memory (item 21), and self-distancing (item 22). Single memories showed lower scores than Multiple and Global memories. For emotional intensity/event (item 19), multiple memories showed higher scores than Single and Global memories.

## 5. Discussion

The subjective experience associated with the remembering of the personal past, namely, its phenomenology, is quite relevant in distinguishing between different kinds of ABMs [7,8,9] and in identifying memory changes related to aging [5,10,11,12,13,14]. Although standardized instruments have been recently developed to provide a comprehensive evaluation of the phenomenology of ABMs, a web-based assessment of the full range of phenomenological properties of ABMs is still missing. In the few online studies, the assessment of the phenomenology has been done by simply creating ad hoc items [58] or using instruments originally developed in paper-and-pencil format, without investigating the psychometric properties of web-based versions (e.g., [4,5,14]).

In the present study, we developed and tested the psychometric properties of a self-administered web-based version of the Assessment of the Phenomenology of Autobiographical Memory (APAM), a 27-item questionnaire that enables to assess a wider range of phenomenological properties compared to other standardized instruments and does it in a quick, non-intrusive, and effective way.

The online version of APAM tested in the present study replicated the good psychometric properties of the original paper-and-pencil version: specifically, all items showed a high consistency of ratings computed across multiple cue words and multiple measurement time points (over a period of 7–10 days) and this pattern of results was observed in young adults and older adults. 

These results suggest that the items of APAM measure the same dimensions regardless of the cue administered and, even more important, that the measurement is consistent over time, regardless of fluctuations in mood and arousal. The replication of these results in young and older adults makes the online version of APAM a promising instrument suitable for an in-depth assessment of the phenomenology in a broad range of samples and contexts. 

One of the advantages of web-based measurement is to increase the availability of testing and participants [31,32]: web-based methodology provides and efficient and cost-effective solution, especially when we are interested in comparing multiple measures taken over time, e.g., several days, as in the present study, or several years longitudinally, but also in different contexts (e.g., measurements administered in clinical/laboratory settings and in the home environment).

Over the last year, an increasing number of web-based cognitive tests have been developed for measuring cognitive function in middle-aged and elderly people in unsupervised settings (e.g., [34,59]). Globally, these studies have shown that older adults rated computer-based and web-based self-administered measures as more user-friendly and less stressful than paper-and-pencil tests [37,38,60]. For example, in the study by Hansen et al. [60], older adults and seniors reported a preference for a web-based self-administered version of the neuropsychological battery Memoro compared to a traditional paper-and-pencil analogue, as it allowed them to set their own pace, and they did not feel scrutinized by a test administrator. Our results confirm that a web-based assessment of the phenomenology of ABMs is feasible also with older adults, and the flexibility of scheduling might contribute to a less taxing experience, increasing compliance and reducing the risk of dropout. In the present study, we also tested for age differences in the phenomenological properties of ABMs. Compared to younger participants, older adults rated their ABMs as more vivid and coherent, and they reported a stronger experience of recollection and reliving (sensory and emotional) and higher confidence in the accuracy of their ABMs.

These results are consistent with previous findings, showing that the subjective experience of ABMs tends to become stronger with age, with older adults reporting higher ratings of phenomenology (e.g., vividness, coherence, reliving) [10,11,12,13,14,58] as well as higher levels of confidence in memory accuracy (e.g., [61]) despite the impairment in the retrieval of episodic details (e.g., [15,16,17,18,19]). In the literature, different explanations have been proposed for the stronger phenomenology among older adults (see for a discussion, [5]). For example, in some cases, when asked to retrieve ABMs, older adults may have more possibilities, compared to young adults, to select ABMs with a higher personal importance and centrality to their identity, and this would make these memories less subject to aging decline and associated with a more intense phenomenology. The richness of the phenomenology may also depend on the higher frequency of rehearsal of ABMs (i.e., think about the past and share ABMs with others) in older adults, which might enhance memory accessibility and reliving experience. Moreover, we cannot exclude that older adults had a response bias, being more likely to report higher scores in the phenomenology ratings. Indeed, our results speak against both a generalized increase in the phenomenology with aging and a response bias, since we found that older adults reported a stronger experience of recollection and reliving (i.e., sensory and emotional), higher vividness and coherence of ABMs as well as a higher confidence in memory accuracy compared to young adults, but we also found that the two groups did not differ in other important characteristics, as the richness of sensory details of ABMs, the intensity of feeling experienced at retrieval and during the original event, and other subjective judgments of some properties of remembered events and memory, as the personal importance of memory, its accessibility, its specificity and the frequency of sharing with other people (talking about it). This pattern of results demonstrates the complexity of the phenomenological structure of memory, and the necessity of a comprehensive assessment to characterize the phenomenological profile of young and older adults. It could be argued that it does not seem a matter of whether the phenomenology declines or increases with aging, but *how* it changes.

In evaluating these results, some limitations, as well as a number of possibilities for future studies, can be identified. The two groups may appear small in absolute size, but it should be considered that the number of data points is seven times higher, as each participant has been tested seven times. Besides, we chose not to try and recruit further participants as this would have likely led to sample from potentially different populations in either group (e.g., non-student participants in the young group, older adults with different motivations to take part in the study). Heterogeneity of populations is a known biasing factor for correlation-based analyses, and pooling data from different samples to obtain better estimates with more observations is a procedure usually advised against (e.g., [62]). Furthermore, the sampling strategy could have been a protective factor against dropping out from the study, together with the online administration method.

The downside of this, however, was that our participants were all highly educated, and our sample of older adults consisted of active and intellectually stimulated retired people, thereby raising questions about the generalizability of our findings. Future studies with young and older adults representative of the population they are intended to reflect are needed to empirically address this issue.

Although the results presented in this study allow a comparison of the web-based administration of APAM with the paper-and-pencil one reported in Vannucci et al. [6], a more principled method for comparing the two versions would be to administer both of them to the same group of participants. For instance, an even number of cues could be chosen, and a random half of them could be administered in a laboratory setting followed by the administration of the paper-and-pencil APAM, and the other random half could be administered online. By appropriately randomizing the cues within participants and between administration methods, it would be possible to reliably compare the effects of the cues and of the administration method. Other between factors, such as age group, could also be considered.

Another promising application of APAM would be adapting the items to provide an assessment of thinking, and not just memories. In this case, APAM would allow a within-subject design to assess and compare the phenomenology of past and future thinking in both young and older adults. In this regard, a recent study by El Haj and Lenoble [63] with young adults demonstrated more subjective vividness of past than future thinking. However, in the study by El Haj, Antoine and Kapogiannis [64], (see also [65]) carried out in patients with Alzheimer’s disease and older controls, there were no significant differences in the subjective experience of reliving during the generation of past and future events, thereby suggesting possible changes in the phenomenological experience associated with past and future thinking with aging.

Keeping with this, another potential extension of this study is the application of online APAM to other populations of special interest for research on ABMs, such as adolescents and clinical patients. Adolescents are technologically savvy and often prefer to answer questions using a web-based platform compared to the standard paper-and-pencil assessment. Recent studies have already shown that adolescents can accurately and reliably fill in web-based questionnaires about their mental and physical well-being under an unsupervised condition (e.g., [66,67,68]).

Given its brevity and ease of administration online, the APAM could be also a useful add-on to assessment batteries for psychological problems. Alterations in the phenomenology of ABMs have been consistently reported in depression [69,70] and post-traumatic stress disorder (e.g., [71,72]). In a recent study, Ashbaugh, Marinos, and Bujaki [73] investigated how symptoms of PTSD and depression influence the phenomenological properties of trauma memories. Their results show that the symptoms of PTSD and depression were related separately and uniquely to the phenomenology of trauma memory, demonstrating that depressive and PTSD symptomatology affect traumatic memory differently. In this regard, the application of the online APAM to comprehensively assess the phenomenology of different kinds of ABMs (e.g., trauma vs non-trauma memories) in clinical patients might help clarify the question, of theoretical and clinical interest, of the complex mechanisms of the influence of psychopathology on ABMs.

## 6. Conclusions

The self-administered web-based version of APAM tested in the present study replicated the good psychometric properties of the original paper-and-pencil version. Results revealed an adequate level of consistency of ratings computed across multiple cues and multiple measurement time points in both young and older adults. This pattern of results makes the online version of APAM a versatile instrument suitable for a comprehensive assessment of the phenomenology in a broad range of samples and contexts.

## Figures and Tables

**Table 1 brainsci-11-00660-t001:** Descriptive statistics and rating consistency for the APAM items in young and older adults.

	Young (*n* = 35)			Older Adults (*n* = 45)				
Item	M	SD	ICC (95% CI)	M	SD	ICC (95% CI)	adj. *p*	r
item01-clarity	4.67	1.22	0.82 (0.70; 0.90)	5.29	0.93	0.70 (0.54; 0.81)	0.303	0.13 (−0.08; 0.33)
item02-color	6.13	0.72	0.69 (0.50; 0.83)	5.51	1.03	0.73 (0.59; 0.84)	0.738	0.04 (−0.21; 0.27)
item03-vividness	4.50	1.22	0.87 (0.78; 0.92)	5.18	0.90	0.71 (0.56; 0.82)	0.113	0.20 (0.02; 0.38)
item04-richness of visual details	6.01	0.58	0.75 (0.60; 0.86)	5.63	0.79	0.81 (0.71; 0.88)	0.575	0.07 (−0.16; 0.33)
item05-sound	4.06	1.27	0.79 (0.67; 0.88)	3.92	1.00	0.68 (0.51; 0.80)	0.368	0.12 (−0.11; 0.35)
item06-smell	2.83	1.16	0.74 (0.58; 0.85)	2.96	1.24	0.82 (0.73; 0.89)	0.419	0.10 (−0.10; 0.30)
item07-touch	3.18	1.36	0.81 (0.69; 0.89)	3.25	1.10	0.67 (0.51; 0.80)	0.303	0.14 (−0.06; 0.34)
item08-taste	2.39	1.22	0.83 (0.72; 00.90)	2.38	0.98	0.69 (0.52; 0.81)	0.246	0.16 (−0.10; 0.39)
item09-sensory reliving	4.35	1.18	0.84 (0.74; 0.91)	4.60	0.83	0.67 (0.50; 0.80)	0.135	0.19 (−0.02; 0.40)
item10-auditory reliving	3.94	1.31	0.84 (0.74; 0.91)	4.05	0.90	0.65 (0.47; 0.79)	0.113	0.21 (−0.00; 0.41)
item11-visual reliving	5.03	0.86	0.78 (0.64; 0.87)	4.94	0.76	0.66 (0.48; 0.79)	0.390	0.12 (−0.06; 0.30)
item12-spatial reliving	4.47	0.99	0.79 (0.66; 0.88)	4.57	0.89	0.75 (0.62; 0.85)	0.687	0.05 (−0.11; 0.21)
item13-remember/know	5.64	0.73	0.62 (0.38; 0.79)	5.91	0.84	0.86 (0.78; 0.91)	0.050	0.27 (0.03; 0.49)
item14-formulation in words	3.17	1.66	0.94 (0.90; 0.97)	3.69	1.15	0.81 (0.71; 0.89)	0.025	0.28 (0.10; 0.48)
item15-coherence	3.92	1.36	0.86 (0.78; 0.92)	4.88	0.95	0.77 (0.65; 0.86)	0.303	0.13 (−0.06; 0.32)
item16-confidence in the accuracy of the memory/testify	5.30	1.00	0.78 (0.65; 0.88)	5.64	1.11	0.87 (0.80; 0.92)	0.303	0.13 (−0.14; 0.36)
item17-accessibility	5.27	0.99	0.70 (0.51; 0.83)	5.37	0.70	0.62 (0.43; 0.77)	0.589	0.07 (−0.16; 0.31)
item18-visual perspective/third person	3.30	1.53	0.86 (0.77; 0.92)	3.61	1.35	0.84 (0.75; 0.90)	0.738	0.03 (−0.17; 0.21)
item19-emotional intensity/event	5.14	0.78	0.68 (0.48; 0.82)	5.33	0.73	0.77 (0.64; 0.86)	0.518	0.09 (−0.12; 0.29)
item20-emotional reliving of the event	4.57	0.91	0.69 (0.50; 0.83)	4.85	0.84	0.77 (0.65; 0.86)	0.543	0.08 (−0.12; 0.27)
item21-emotional intensity/memory	4.49	1.10	0.77 (0.63; 0.87)	4.52	0.90	0.78 (0.66; 0.86)	0.968	0.00 (−0.20; 0.24)
item22-self-distancing	4.25	1.44	0.82 (0.71; 0.90)	4.47	1.15	0.74 (0.61; 0.84)	0.419	0.09 (−0.14; 0.30)
item23-rehearsal of memories/talking	3.31	1.28	0.77 (0.63; 0.87)	3.05	0.93	0.66 (0.48; 0.79)	0.419	0.11 (−0.14; 0.32)
item24-personal importance	3.60	1.33	0.79 (0.66; 0.88)	3.94	0.95	0.70 (0.54; 0.81)	0.419	0.10 (−0.08; 0.30)
item25-imagined/real	6.21	0.70	0.81 (0.70; 0.89)	6.55	0.46	0.79 (0.68; 0.87)	0.738	0.03 (−0.16; 0.17)
item26-Specificity (proportions)							0.374 °	0.06 (−0.02; 0.12) ^§^
Single Memory (event that occurred once within a single day and place)	0.62			0.65				
Multiple Memory (summary or merging of similar events)	0.25			0.20				
Global Memory (event that extended for a period greater than on day)	0.13			0.15				
item27-Age of memory (proportions)							<0.001 °	2.53 (1.93; 3.12) ^
One week	0.04			0.00				
One month	0.02			0.01				
Six months	0.10			0.02				
One year	0.13			0.04				
2–5 years	0.32			0.07				
6–10 years	0.19			0.04				
11–15 years	0.11			0.08				
16–30 years	0.07			0.17				
31–50 years	0.00			0.34				
+50 years	0.00			0.22				

Note: M: mean score; SD: standard deviation; ICC: Intraclass Correlation Coefficient; bracketed values indicate the lower and upper limit, respectively, of the 95% confidence interval for the ICC. adj-*p*: *p*-values adjusted for false discovery rate using the Benjamini-Hochberg [52] procedure; *r*: effect size in *r* metric (thresholds for small, moderate, and large effect sizes are 0.10, 0.30, and 0.50, respectively, as in Cohen [53]); °: not included in the correction for false discovery rate; ^§^: Cramer’s *V*; ^: Cohen’s *d*, see text for the results of this comparison.

## Data Availability

The data presented in this study are available on request from the corresponding author. The data are not publicly available as stated in the informed consent form that participants signed.

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
