# Peer review of "Web-Based Assessment of the Phenomenology of Autobiographical Memories in Young and Older Adults"

_brainsci, 2021, doi:10.3390/brainsci11050660_

Round 1

Reviewer 1 Report

Authors did a good job revising the ms. All my queries were answered, and I think the clarity of the ms has improved.

Please add the name of the authors line 107.

Author Response

As the Reviewer suggested, we added  the name of the authors at line 107.

Reviewer 2 Report

I would like to thank the authors for addressing my questions. The methods are much clearer now. I only have a few comments/suggestions:

I think it should be specifically acknowledged in the discussion that the population you tested is unique in that they are highly educated and active older adults, and therefore generalizability remains to be determined. There are pros and cons to using a population like this and I realize this is a pervasive issue in many studies, but you haven’t actually acknowledged it as a limitation here (beyond stating that heterogeneity can be problematic, but that really depends on your goal doesn’t it?). I think this is particularly important to address for a study like this, given that it was conducted online and going forward may be used with broader populations.

In terms of comparing your data to the paper and pencil test- I believe that the fact that scores are consistent across days actually renders the datasets amenable to direct comparison. Especially if phenomenology can be considered as a stable trait, as you point out. However, I will let Reviewer 1 decide if this is worth pursuing as your reframing of the paper helps alleviate the notion that your intent was to answer this question. But surely this is something to think about.

There are still some “elderly” terms scattered throughout, a control F would help you find them all. I noticed some typos throughout as well, so good to double check that.

Author Response

Rev: I think it should be specifically acknowledged in the discussion that the population you tested is unique in that they are highly educated and active older adults, and therefore generalizability remains to be determined. There are pros and cons to using a population like this and I realize this is a pervasive issue in many studies, but you haven’t actually acknowledged it as a limitation here (beyond stating that heterogeneity can be problematic, but that really depends on your goal doesn’t it?). I think this is particularly important to address for a study like this, given that it was conducted online and going forward may be used with broader populations.

We agree with the reviewer. In the revised version (line 483-487) we say that:

The downside of this, however, was that our participants were all highly educated and our sample of older adults consisted of active and intellectually stimulated retired people, thereby raising questions about the generalizability of our findings. Future studies with young and older adults representative of the population they are intended to reflect are needed to empirically address this issue.

Rev:In terms of comparing your data to the paper and pencil test- I believe that the fact that scores are consistent across days actually renders the datasets amenable to direct comparison. Especially if phenomenology can be considered as a stable trait, as you point out. However, I will let Reviewer 1 decide if this is worth pursuing as your reframing of the paper helps alleviate the notion that your intent was to answer this question. But surely this is something to think about.

We agree with the reviewer that this is something to think about but, as the reviewer noticed, our study was not aimed at addressing this issue. We think that this issue needs to be empirically addressed in a future study, and for this reason we did not discuss it in the present work. 

Rev: There are still some “elderly” terms scattered throughout, a control F would help you find them all. I noticed some typos throughout as well, so good to double check that.

We double checked. 

This manuscript is a resubmission of an earlier submission. The following is a list of the peer review reports and author responses from that submission.

Round 1

Reviewer 1 Report

This study is timely and raises relevant questions for current research on autobiographical memory. Thank you for giving me the opportunity to read and review this manuscript.

I think this paper is worth publishing but I have few comments to help to improve the proper understanding of the manuscript, in particular the main aim and focus of the study.

Introduction

In the introduction, authors might want to also mention the ART (Autobiographical Recollection Test; Berntsen, Hoyle and Rubin, 2019) as it is a recent scale developed to measure individual differences in autobiographical memory remembering. This scale also includes measures of phenomenological aspects of autobiographical recollection. The work presented in Berntsen, Hoyle and Rubin, 2019 also emphasized the stable characteristics of autobiographical recollection in individuals, so do the authors in the present ms.

I was a bit confused while reading the aims of the current study. The authors should introduce more precisely which psychometric properties of the web-based version of APAM have been tested in the present study. As a secondary aim, authors mentioned the test of consistency of APAM within and between participants, so what was the first aim? This part needs clarification.

Similarly, in the discussion the authors seem to say that the aim of the study was to address a gap which was “a number of studies have shown that the psychometric equivalence (e.g., reliability, factor structure, and construct validity) between paper-and-pencil and online versions of the same instrument must be demonstrated rather than assumed on the basis of those of the original version.”. But this gap was not presented as being the main aim of the study nor really addressed by the study.

Authors might want to compare some of their results with the ones previously obtained with paper and pencil to really answer this gap.

Moreover, as it is presented, it is confusing for the reader whether the authors were interested in assessing the scale reliability – consistency of measurement as a psychometric property of the scale – or the stability of phenomenological characteristics of autobiographical memory in individuals, as a basic research question.

Method

Authors mentioned a “newly developed web-based version of APAM”, however it is not clear in the method section which parts of the APAM have been adapted for web experiments.

In the APAM questionnaire, I wonder whether items 23, 24, 25; 26, 27 can really be considered as phenomenological properties of autobiographical memory.

The authors have taken into account several confounding variables when computing the analyses. I was wondering whether it was possible to control for personal importance of the memory when assessing consistency of ratings within participants?

Discussion

Lines 428-429 – is that because not all items of APAM actually assess phenomenological properties?

I’m not sure to understand the interpretation made lines 433-437. How do the authors discriminate component processes from phenomenological state, from properties of recalled autobiographical memory? I suggest clarifying this point.  

Reviewer 2 Report

The authors aimed to test the psychometric properties of the web-based version of the APAM, which is a paper and pencil test that assesses the subjective experience associated with retrieving autobiographical memories. The authors administered the online APAM to a group of young and old participants, and examined rating consistency of scores elicited by each cue (administered on separate days) within and across groups, as well as group differences in phenomenological ratings for each item. They found the groups were mostly consistent in their responses to each cue across days, except for the “formulation in words” item (older adults were less consistent) and the “remember/know” item where younger adults were less consistent. They then went on to show that older and younger adults differed considerably in their phenomenological experience on a number of items. Notably, older adults tended to have higher phenomenological scores along several dimensions.

Creating and validating web-based tests is especially important in the age of COVID-19 when many people are increasingly relying on such tools for research. The test they describe is comprehensive and assesses multiple dimensions of phenomenology. The fact that they presented each cue on different days was a clever way to assess consistency within and across subjects as arousal and mood varies over time. I appreciated that the authors corrected for multiple comparisons where appropriate to reduce the likelihood of false positives.

My questions and concerns are as follows:

  1. I found the methods to be vague (although I did appreciate the authors including the full APAM in the supplementary material). What platform was the online test hosted on, or was it an independent website (in which case is there a link to it that the reader could check out?). Were questions given one at a time or did they see all of the questions at once? What statistical software and/or packages were used?

  1. The authors provided little description of the statistical tests used throughout the paper. Sometimes it was unclear which test was chosen and why, which independent variables were used, in the case of mixed models, which variables were fixed effects vs random variables, and did you use a random intercept or slope? Which if any variables were covariates of no interest? This should be specified throughout, for clarity and for replication purposes.

  1. I find the choice to only test people in psychology courses to be a little bit strange given that a big benefit of web-based testing is that it can be more inclusive of diverse populations than paper and pencil testing. I get that this is a well-controlled population and why that is desirable, but if this test is going to be used online one would think the populations it will actually be used with will be quite different (for example, researchers may take this to MTurk or Prolific)?. How valid will this test really be with a more representative population? Having a highly educated elderly population that is actively in school seems especially niche.

  1. The cues used to elicit memories are also limiting when you are considering a broader application of this test. As someone who lives nowhere near the sea or mountains, my first thought was that participants with low SES would be at a disadvantage, given that they may not have personally experienced the sea or mountains, depending on where they live. Are the same cues always given with this test, or is there flexibility in which cues can be used? Do the specific cues effect the type of memories elicited/has this been tested?

  1. Was there a statistically significant difference in the number of memories elicited by the cue words between the groups? I see descriptive info on this but not a direct comparison.

  1. I see that you did not test for group differences in demographic variables until later on in the results section. I think it is more customary to do this first in the section where you describe your participants, and can then account for differences in demographics throughout the subsequent analyses. Because the authors did not do this, it is hard to interpret some of their early results, because they subsequently show a group difference in education and it is unclear if this may explain earlier observed group differences in other measures. Does controlling for education change any of the results regarding group differences in consistency scores? Does it change the lack of group differences observed for items 24 and 26 regarding significance and specificity of the memories? It looks like based on supplemental table 2 the results for item 24 do not change, but it would be helpful to mention in the main text. Since education differs between groups, it should be controlled throughout.

I am also confused because I believe you corrected for education and age of memory before running the models in supp Table 2, why did you continue to include age of memory in these models but not education?

  1. The finding that older adults have stronger phenomenological score along many dimensions was actually quite surprising to me, and I think warrants some discussion. For example, there are well described differences in quality of memory in older vs younger adults, such that older adults’ memories tend to lack episodic detail, and they tend to rely more on familiarity than recollection. But your results with this measure shows that older adults have more vivid memories with more sensory detail, and are more likely to “remember” than “know”. What accounts for the discrepancy in these two lines of research? Is it because your test is measuring more subjective than objective memory (i.e. older adults are bad at reporting these things)? Could it have to do with response bias? Are older adults more likely to say all memories are more vivid for example, regardless of age of the memory and significance of the memory?

  1. This study really don’t tell us how this online test compares to the presumably already validated paper and pencil test. The authors rightly point out that it would be better to have people do the online and paper and pencil tests to directly compare. I see you have published with this test before, can you not compare to old data to actually answer this question?

Minor

  • It would be helpful to have p values in table 1 in the manuscript for easy comparison (I see you have it in supplemental but would be easier if in the main text)
  • I believe “sex” is a more acceptable term than “gender” these days, assuming you mean biological sex.
  • “elderly” seems a little cruel for people in their mid 60s, I prefer “older adults”. But I suppose that is a preference.